# Confident Block Diagonal Structure-Aware Invariable Graph Completion for Incomplete Multi-view Clustering

**Shuping Zhao[1], Yulong Chen[2], Jie Wen[*2], Lunke Fei[1], Jinrong Cui[3], Tingting Chai[4]**

[1] Guangdong University of Technology  [2] Harbin Institute of Technology, Shenzhen
[3] South China Agricultural University  [4] Harbin Institute of Technology
yb77458@um.edu.mo, jiewen_pr@126.com,

## Abstract

Multi-view clustering (MVC) adopts complementary information from multiple views to reveal the underlying structure of the data. However, the conventional MVC-based methods remain a crucial challenge on the incomplete multi-view clustering (IMVC) tasks, when some views of the multi-view data are missing. Particularly, current IMVC methods suffer from two main limitations: 1) they focused on recovering the missing data, yet often overlooked the potential inaccuracies in imputed values caused by the absence of true label information; 2) the recovered features were learned from the complete data, neglecting the distributional discrepancy between the complete and incomplete instances. In order to tackle these issues, in this paper, a confident block diagonal structure-aware invariable graph completion-based incomplete multi-view clustering method (CBDS_IMVC) is proposed. Specifically, we first design a confident-aware missing-view inferring strategy, where the confident block diagonal structures (CBDS) are learned to guarantee that recovered instances of all views have the same strict invariable local structure with the constraint of CBDS. Subsequently, we proposed an invariable graph completion strategy to learn the intrinsic structure across all views. Each parts are jointly trained, complementing and promoting each other to achieve the optimum together. Compared to other state-of-the-art methods, the proposed CBDS_IMVC demonstrates superior performance across multiple benchmark datasets.

## 1 Introduction

With the rapid development of multi media and sensor technologies, describing a single object from different data sources or modalities has become increasingly convenient Wong et al. (2025); Fang et al. (2023); Wang et al. (2025). For example, a web page can be described by its textual content and accompanying images;the same news might be written in different languages; a facial image may be captured as a photo, infrared image, or sketch Deng et al. (2023); Wang et al. (2021a); Chen et al. (2025). These diverse representations, referring to the same entity, constitute what is known as *multi-view data* Wang et al. (2021b).

Each view in multi-view data corresponds to a distinct perspective, feature space, or data source. These views often exhibit both consensus (shared) and complementary (unique) information Moujahid & Dornaika (2025); Dornaika & El Hajjar (2024). The task of clustering such multi-view data into meaningful groups is known as *Multi-View Clustering (MVC)* Wang et al. (2022). MVC has been extensively studied and widely applied in recent years Fang et al. (2023). For example, Chen et al. Chen et al. (2021) developed a low-rank representation-based method to enforce global structure consistency across views. Li et al. Li et al. (2021) proposed a spectral consensus strategy by minimizing the disagreement among Laplacians from each view, leading to a more robust clustering

---

*Corresponding Author: Jie Wen, e-mail: jiewen_pr@126.com.

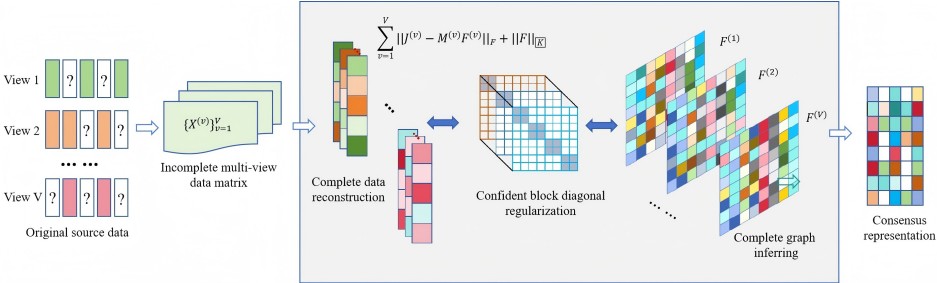

Figure 1: The flowchart of the proposed CBDS_IMVC method.

performance. In addition, Tang et al. Tang et al. (2023) introduced a subspace clustering method via adaptive graph learning and late fusion alignment.

While this approach has shown promise in some scenarios, it suffers from lacking of robustness to noisy views for it equally treatment of all available views ignores their reliability disparitiesWen et al. (2021). Graph-based multi-view clustering methods typically construct similarity graphs for each view using metrics like Gaussian kernels or cosine similarity, then fuse view-specific graphs through linear combination or advanced tensor operations, and perform spectral clustering on the unified graphTang et al. (2023); Wang et al. (2019a). However, these methods are sensitive to the choice of similarity measure. Additionally, they are prone to performance degradation when data is sparse or missing, as incomplete or noisy graphs can hinder clustering accuracy. Deep learning-based methods have also gained popularity due to their ability to learn complex nonlinear relationships and handle missing data through specialized architectures Du et al. (2021); Wang et al. (2019b); Chowdhury et al. (2025). Despite their flexibility, deep learning methods often require large amounts of training data and extensive hyperparameter tuning, limiting their applicability in resource-constrained scenarios.

To sum up, most IMVC methods focus on learning a common graph from the available views, neglecting the latent information embedded in the missing views Cui et al. (2024). In order to tackle these issues, in this paper, a confident block diagonal structure-aware invariable graph completion-based incomplete multi-view clustering method (CBDS_IMVC) is proposed. As shown in Figure 1, we first propose a symmetric-aware missing-view inferring strategy, where the confident block diagonal structures (CBDS) are designed to guarantee that the recovered instances of all views have the same strict invariable local structure with the constraint of CBDS. Subsequently, we proposed an invariable graph completion term to learn the intrinsic structure across all views. This ensured that the reconstructed data accurately captured the true distribution of missing instances in each view. The main contributions is summarized as follows:

1) Different from the other related papers, the proposed CBDS_IMVC designed a complete representation reconstruction strategy, where the consensus complete structure across all views was learned to preserve the structure of the reconstructed data.

2) A novel confident block diagonal regularizer is proposed to guide all views toward learning an invariable structure with block diagonal form. This mechanism preserves distribution of the data from each view while enforcing a consistent and strict block diagonal pattern that remains aligned across all views.

3) Extensive experiments demonstrate that the proposed CBDS_IMVC can obtain the best performances compared with the other state-of-the-art methods.

## 2 THE PROPOSED METHOD

### 2.1 SYMMETRIC-AWARE MISSING-VIEW INFERRING

Let $\{X^{(v)} \in \mathbb{R}^{d^{(v)} \times n^{(v)}}\}_{v=1}^{V}$ represent the incomplete multi-view data, where $V$ is the total number of views and $n$ denotes the number of subjects. Here, $d^{(v)}$ and $n^{(v)}$ correspond to the feature dimen-

sionality and the number of the available samples in the $v$-th view, respectively. Given the affinity graph $S^{(v)}$ derived from observed instances in each view, we can construct its complete graph counterpart $\widetilde{S}^{(v)}$ by padding zeros for all missing instances. After obtaining the initial extended complete graph $\widetilde{S}^{(v)}$ for each view, we proposed the missing data referring strategy as follows:

$$\min_{J^{(v)}, M^{(v)}} \sum_{v=1}^{V} ||J^{(v)} - M^{(v)}\widetilde{S}^{(v)}||_F^2 \ s.t. \ \mathcal{P}o_v(M^{(v)}\widetilde{S}^{(v)}) = \mathcal{P}o_v(\widetilde{X}) \tag{1}$$

where $\{J^{(v)} \in \mathbb{R}^{d^{(v)} \times n}\}_{v=1}^{V}$ is the reconstructed complete samples from all views, $M^{(v)}$ denotes the reconstructed complete data matrix for the $v$-th view. The operation $M^{(v)}\widetilde{S}^{(v)}$ ensures that the geometric structure of available instances is maintained, subject to the topological constraints imposed by the complete graph $\widetilde{S}$. Additionally, $\mathcal{P}o_v$ denotes the projection matrix associated with view $v$, which is used to record the existence of instances in the dataset $X^{(v)}$. Specifically, it operates through one-hot encoding, where a value of 1 in a row signifies the presence of a specific instance in view $v$, while a value of 0 denotes its absence.

Specifically, $\mathcal{P}o_v$ takes a value of 1 if the instance $X_i^{(v)}$ exists and 0 if it does not exist. Mathematically, the extension of $\widetilde{S}^{(v)}$ can be transformed as follows:

$$\widetilde{S}^{(v)} = G^{(v)} S^{(v)} G^{(v)^T} \tag{2}$$

where $s_{i,j}^{(v)} = e^{-\frac{||x_i^{(v)} - x_j^{(v)}||^2}{2}}$ is a Gaussian kernel function to present the symmetric relationships, $G^{(v)} \in$
$\mathbb{R}^{n_v \times n}$ serves as a projection matrix that maps the incomplete structure of the $v$-th view to its complete extended version. Specifically, for the $j$-th instance $x_j^{(v)}$ in $X^{(v)}$, we define $G^{(v)}$ as follows:

$$G_{i,j}^{(v)} = \begin{cases} 1, & if \ x_j^{(v)} \ is \ the \ vth \ view \ of \ the \ ith \ sample \\ 0, & otherwise. \end{cases} \tag{3}$$

## 2.2 Confident Block Diagonal Structure Preserving

In order to let the referring instances preserve the data distribution of the original data, all views should maintain a as similar data structure as possible. Hence, we propose the symmetric-aware missing-view inferring strategy, which adopts a block diagonal structure constraint to maintain the same structure for the reconstructed data of all views as follows:

$$\min_{J^{(v)}, M^{(v)}, F^{(v)}} \sum_{v=1}^{V} ||J^{(v)} - M^{(v)}F^{(v)}||_F^2 + ||F||_{\boxed{k}} \ s.t. \ \mathcal{P}o_v(M^{(v)}F^{(v)}) = \mathcal{P}o_v(\widetilde{X}) \tag{4}$$

Seeking to precisely reveal and preserve the structural distribution of the reconstructed complete data for all views, we propose the confident block diagonal structure induced regularizer $||F||_{\boxed{k}}$. For any affinity matrix $F^{(v)}$, its block diagonal regularizer $||F^{(v)}||_{\boxed{k}}$ is defined as follows:

$$\| F^{(v)} \|_{\boxed{k}} = \| F \|_{\circledast} + \lambda \| F \|_1 \tag{5}$$

where $\lambda$ is a nonnegative penalty parameter, $\| F \|_1$ is a close-form to $\| Z \|_0$, which is a NP-Hard problem. Here, $\| F^{(v)} \|_{\boxed{k}}$ is the proposed confident block diagonal regularization on multiple views. Specifically, $F^{(v)}$ captures complete structural information for both missing and available instances in each view by adopting the self-representation property of the data with $\ell_1$ regularization. This approach enables selective incorporation of the most reliable information from other views to achieve structure completion. What is more, the tensor low-rank constraint $\| \mathcal{Z} \|_{\circledast}$ effectively captures high-order correlations across multiple views, serving as a crucial component in our framework.

However, $||F||_{\boxed{k}}$ still fails to account for the latent information of missing instances in each view. To address this limitation, we propose an intrinsic complete structure inference strategy, which effectively adopts complementary information across different views to recover the complete intrinsic structure of each view. Consequently, we jointly optimize the referring of the complete multi-view data and the reconstruction of the confident complete graph as follows:

$$\min_{\phi} \sum_{v=1}^{V} ||J^{(v)} - M^{(v)}F^{(v)}||_F^2 + \lambda||F||_{\boxed{k}} + \sum_{v=1}^{V} \lambda_1 ||(F^{(v)} - \sum_{i=1,i\neq v}^{V} \widetilde{S}^{(i)}B_{i,v}) \odot E^{(v)}||_F^2$$

$$s.t.\ \mathcal{P}o_v(M^{(v)}F^{(v)}) = \mathcal{P}o_v(\widetilde{X}), 0 \leq B_{i,v} \leq 1,\ \sum_{i=1,i\neq v}^{V} B_{i,v} = 1,\ B_{v,v} = 0 \tag{6}$$

where $\phi = \{J^{(v)}, M^{(v)}, F^{(v)}, B\}$, $\lambda_1$ is a nonnegative penalty parameter, $B \in R^{n\times n}$ is the linear representation matrix, $\odot$ is the element-wise multiplication to preserve the real structure of the available instances, and $E^{(v)} = Q_{v,:}^T Q_{v,:}$ indicates whether both the $i$-th and $j$-th samples have instances in the $v$-th view. Specifically, $Q^{(v)} \in R^{V\times n}$ represents the missing-view indicator matrix, where $Q_{i,j} = 0$ means the $j$-th instance is missing from the $i$-th view, while $Q_{i,j} = 1$ indicates its presence.

## 2.3 FINAL OBJECTIVE FUNCTION

The complete structure inference enables us to recover the intrinsic structure of the latent complete data for each view. To complement the main objective and enhance the clustering performance, a consensus learning module is introduced to obtain a unified low-dimensional representation shared across all views for spectral clustering. In this work, we unify these three components through a joint learning framework to achieve optimal performance. The complete objective function is formulated as follows:

$$\min_{\Phi} \sum_{v=1}^{V} (\alpha^{(v)})^r \left( ||J^{(v)} - M^{(v)}F^{(v)}||_F^2 + \lambda||F||_{\boxed{k}} + \lambda_1 ||(F^{(v)} - Y^{(v)}) \odot E^{(v)}||_F^2 \right)$$

$$+ \sum_{v=1}^{V} (\alpha^{(v)})^r \lambda_2 Tr(U^T L_{Y^{(v)}} U)\ s.t.\ \mathcal{P}o_v(M^{(v)}F^{(v)}) = \mathcal{P}o_v(\widetilde{X}),\ U^{(v)T}U^{(v)} = I,$$

$$0 \leq B_{i,v} \leq 1,\ \sum_{i=1,i\neq v}^{V} B_{i,v} = 1,\ B_{v,v} = 0, \tag{7}$$

$$Y^{(v)} = \sum_{i=1,i\neq v}^{V} \widetilde{S}^{(i)}B_{i,v},\ diag(F^{(v)}) = 0,\ F^{(v)} \geq 0,\ F^{(v)} = F^{(v)T}$$

where $\Phi = \{J^{(v)}, M^{(v)}, F^{(v)}, B, Y^{(v)}\}$, $\lambda_2 \geq 0$ presents a penalty parameter, $U \in R^{n\times c}$ denotes the consensus representation, and $\alpha^{(v)}$ represents the weighting factor for balancing missing instances in each view, with higher values indicating greater importance of the corresponding view. Here, $L_{Y^{(v)}}$ denotes the Laplacian matrix of the complete structure $Y^{(v)}$, computed as $L_{Y^{(v)}} = D^{(v)} - \frac{(Y^{(v)}+Y^{(v)})}{2}$. Specifically, $D^{(v)}$ is a diagonal matrix whose $i$-th diagonal element is given by $D^{(v)}i, i = \sum_{j=1}^{V} \frac{Y^{(v)}i,j + Y^{(v)}j,i}{2}$.

## 2.4 THE OPTIMAL SOLUTION

In this subsection, we employ the Alternating Direction Method of Multipliers (ADMM) to solve the optimization problem.

**Updated $Y^{(v)}$ and $B$:** By fixing the other variables, $Y^{(v)}$ can be obtained by solving the following problem:

$$\min_{Y^{(v)}} \sum_{v=1}^{V} (\alpha^{(v)})^r \Big( \lambda_1 ||(F^{(v)} - Y^{(v)}) \odot E^{(v)}||_F^2 + \frac{\lambda_2}{2} \sum_{i,j}^{n} K_{i,j} Y_{i,j}^{(v)} \Big)$$

$$+ \sum_{v=1}^{V} (\alpha^{(v)})^r \Big( \sum_{i=1,i\neq v}^{V} B_{v,i}^2 \frac{\mu}{2} ||Y^{(v)} + \frac{C_3^{(v)}}{\mu} - \frac{N^{(i)}}{B_{v,i}}||_F^2 + \frac{\mu}{2} ||Y^{(v)} - M^{(v)}||_F^2 \Big). \tag{8}$$

where $K_{i,j} = ||u_{i,:} - u_{j,:}||_2^2$, $N^{(i)} = Y^{(i)} - \sum_{j=1,j\neq v,j\neq i}^{V} \widetilde{S}^{(j)} B_{j,i}$, and $M^{(i)} = \sum_{i=1,i\neq v}^{V} \widetilde{S}^{(i)} B_{i,v} - \frac{C_3^{(v)}}{\mu}$. Problem (8) can be optimized by solving the following minimization problem:

$$\min_{Y^{(v)}} \sum_{i,j}^{n} ||Y_{i,j}^{(v)} - T_{i,j}^{(v)}||^2, \tag{9}$$

in which $T_{i,j}^{(v)} = P_{i,j}^{(v)} / A_{i,j}^{(v)}$, where $P_{i,j}^{(v)} = \sum_{t=1,t\neq v}^{V} (\alpha^{(v)})^r B_{v,t} \frac{\mu}{2} N^r i,j + \frac{\mu}{2} (\alpha^{(v)})^r M_{i,j}^{(v)} + (\alpha^{(v)})^r \Big( F^{(v)} E^{(v)} - K_{i,j} \Big)$ and $A_{i,j}^{(v)} = (\alpha^{(v)})^r E_{i,j}^{(v)} + \sum_{t=1,t\neq v}^{V} (\alpha^{(v)})^r B_{v,t}^2 \frac{\mu}{2} + \frac{\mu}{2} (\alpha^{(v)})^r$.

Particularly, $B$ can be obtained by solving the optimization problem of $\min_{0\neq B_{i,v}\neq 1, \sum_{i,v}^{(V)}=1, B_{v,v}=0} \sum_{v=1}^{V} ||G_1^{(v)} - \sum_{i=1,i\neq v}^{V} \widetilde{S}^{(i)} B_{i,v}||_F^2$, which is a simplex representation based optimization problem, in which $B$ can be obtained via the accelerated projected gradient method.

**Updated $U$:** By fixing the other variables, $U$ can be obtained by solving the following problem:

$$\mathcal{L}(U) = \lambda_2 \sum_{v=1}^{V} Tr(U^T L_{Y^{(v)}} U) \ s.t. \ U^T U = I. \tag{10}$$

Equation (10) constitutes a standard eigenvalue decomposition problem. The consensus representation matrix $U$ can be obtained as $[u_1, u_2, ..., u_c] \in R^{n\times c}$, where $u_i i = 1^c$ are the eigenvectors corresponding to the first $c$ smallest eigenvalues of $\sum^{V} v = 1 L_{Y^{(v)}}$.

**Updated $Z^{(v)}$:** By fixing the other variables, the auxiliary variable $Z^{(v)}$ can be obtained by solving the following problem:

$$\mathcal{L}(Z^{(v)}) = ||\mathcal{Z}||_{\circledast} + \sum_{Z^{(v)T} Z^{(v)}=I} \frac{\mu}{2} ||\mathcal{F} - Z^{(v)} + \frac{C_2^{(v)}}{\mu}||_F^2$$

$$\rightarrow \mathcal{Z} = \min_{\mathcal{Z}} ||\mathcal{Z}||_{\circledast} + \frac{\mu}{2} ||\mathcal{Z} - \mathcal{U} + \frac{C}{\mu}||_F^2 \tag{11}$$

where $C \in R^{n\times n\times V}$ is a tensor collected by all $\{C_2^{(v)}\}_{v=1}^{V}$. Problem (11) is a typical t-SVD based tensor nuclear norm minimization problem and has the following closed-form solution:

$$\mathcal{U} = \hat{\mathcal{U}} \mathcal{K}_{\hat{\mu}}(\mathcal{S}) \mathcal{V}^T \tag{12}$$

where $\mu = n\mu$, $\mathcal{Z} + \mathcal{A}/\mu = \hat{\mathcal{Z}} \mathcal{S} \mathcal{V}^T$ is obtained by the t-SVD operation. $\mathcal{K}_{\hat{\mu}} = \mathcal{S}\mathcal{J}$, where $\mathcal{J} \in R^{n\times l\times n}$ is a diagonal tensor whose diagonal elements in the Fourier domain are expressed as $\mathcal{J}_f(i,i,j) = max(1 - \hat{\mu}/\mathcal{S}_f^{(j)}(i,i), 0)$.

**Updated $J^{(v)}$ and $R^{(v)}$:** By fixing the other variables, $J^{(v)}$ and the auxiliary variable $R^{(v)}$ can be obtained by solving the following problem:

$$\min_{J^{(v)}, \widetilde{X}^{(v)}, R^{(v)}} ||J^{(v)} - R^{(v)}||_F^2 + \frac{\mu}{2} ||R^{(v)} - G_3^{(v)}||_F^2 \ s.t. \ \mathcal{P}o_v(R^{(v)}) = \mathcal{P}o_v(\widetilde{X}). \tag{13}$$

where $G_3^{(v)} = M^{(v)}F^{(v)} - C_1^{(v)}/\mu$. We can respectively attain $E$ and $R^{(v)}$ as follows:

$$J_{i,j}^{(v)} = \frac{G(v)_{i,j}}{\sum_{j=1,r\neq v}^{V} G(r)_{i,j}} L_{i,j}^{(v)} W_{i,j}^{(v)}, \tag{14}$$

where $G(v)$ is the correlation with cosine distance between the $i$th distance and the $j$th distance on the $v$th view, $L^{(v)} = R^{(v)}W^{(v)^T}(W^{(v)}W^{(v)^T})^{-1}$ and $W^{(v)} = (L^{(v)^T}L^{(v)})^{-1}L^{(v)}R^{(v)}$.

$$R^{(v)} = \frac{1}{\lambda_1 + \mu}(\mu G_3^{(v)} + \lambda_1 E) + \mathcal{P}o_v(\widetilde{X}^{(v)} - W), \tag{15}$$

where $G_3^{(v)} = M^{(v)}F^{(v)} - \frac{C_1^{(v)}}{\mu}$.

**Updated $M^{(v)}$ and $F^{(v)}$:** By fixing the other variables, $M^{(v)}$ can be obtained by solving the following problem:

$$\mathcal{L}(M^{(v)}) = \min_{M^{(v)}} \frac{\mu}{2}\|G_4^{(v)} - M^{(v)}F^{(v)}\|_F^2, \tag{16}$$

in which $G_4^{(v)} = R^{(v)} - C_1^{(v)}/\mu$. Afterwards, we can achieve $M^{(v)} = G_4^{(v)}F^{(v)}(F^{(v)}F^{(v)^T})^{-1}$ by setting the derivative $\partial\mathcal{L}(M^{(v)})/\partial M^{(v)} = 0$. Then, we can attain $F^{(v)}$ by solving the following minimization problem:

$$\mathcal{L}(F^{(v)}) = \min_{F^{(v)}} \lambda_1\|(F^{(v)} - Y^{(v)}) \odot E^{(v)}\|_F^2 + \frac{\mu}{2}|G_4^{(v)} - M^{(v)}F^{(v)}\|_F^2 + \frac{\mu}{2}\|F^{(v)} - G_5^{(v)}\|_F^2, \tag{17}$$

where $G_5^{(v)} = Z^{(v)} - \frac{C_2^{(v)}}{\mu}$. By setting the derivative $\partial\mathcal{L}(F^{(v)})/\partial F^{(v)} = 0$, we can obtain $F^{(v)} = (\mu M^{(v)}G_4^{(v)} + \mu G_5^{(v)})(\mu M^{(v)^T}M^{(v)} + \mu I)^{-1} + \lambda_1 Y^{(v)} \odot E^{(v)}$.

**Updated $P^{(v)}$:** By fixing the other variables, $P^{(v)}$ can be obtained by solving the following problem:

$$L(P^{(v)}) = \frac{\mu}{2}\|F^{(v)} - P^{(v)} + \frac{C_4^{(v)}}{\mu}\|_F^2 + \lambda\|P^{(v)}\|_1 \tag{18}$$

Then, we can obtain the optimal solution of $P^{(v)}$ as follows:

$$P^{(v)} = \Omega_{\frac{\lambda}{\mu}}\left(F^{(v)} + \frac{C_2^{(v)}}{\mu}\right). \tag{19}$$

where $\Omega$ presents the shrinkage operator.

Let $\omega^{(v)}$ denote (7), we can achieve $\alpha^{(v)}$ as $\alpha^{(v)} = (\omega^{(v)})^{1/(1-r)}/\sum_{v=1}^{V}(\omega^{(v)})^{1/(1-r)}$. In addition, the Lagrange multipliers $C_1^{(v)}, C_2^{(v)}, C_3^{(v)}$, and $C_4^{(v)}$ can be updated as $C_1^{(v)} = C_1^{(v)} + \mu(R^{(v)} - M^{(v)}F^{(v)})$, $C_2^{(v)} = C_2^{(v)} + \mu(F^{(v)} - Z^{(v)})$, $C_3^{(v)} = C_3^{(v)} + \mu(Y^{(v)} - \sum_{i=1,i\neq v}^{V} \widetilde{S}^{(i)}B_{i,v})$, and $C_4^{(v)} = C_4^{(v)} + \mu(F^{(v)} - P^{(v)})$, respectively. Particularly, $\mu$ can be updated as $\mu = \min(\rho\mu, \mu_{max})$, where $\rho$ and $\mu_{max}$ are two constants.

## 2.5 COMPUTATIONAL COMPLEXITY ANALYSIS

This paper employs an iterative optimization scheme to solve each variable in the objective function. The update procedures for $Y^{(v)}$ and $B$ involve only simple element-wise operations, rendering their computational cost negligible. The most expensive step is updating $u$, which requires an eigenvalue decomposition of $O(cn^2)$ with the first $c$ minimum eigenvalues. In the step of $Z^{(v)}$,

Table 1: ACC (%), NMI (%), and Purity (%) of different methods on the BBCSport, COIL-20, Caltech-7, and BUAA datasets, respectively.

| Data | Methods | ACC (%) | | NMI (%) | | Purity (%) | |
|---|---|---|---|---|---|---|---|
| | | 0.3 | 0.5 | 0.3 | 0.5 | 0.3 | 0.5 |
| BBCSport | MVL_IV | 72.58±0.50 | 53.45±1.49 | 59.09±5.05 | 38.85±1.36 | 80.75±1.32 | 62.93±1.49 |
| | AWIMVC | 78.33±2.25 | 64.75±2.69 | 69.51±2.17 | 49.48±2.03 | 85.42±0.97 | 73.55±1.20 |
| | UEAF | 68.62±5.87 | 62.93±5.69 | 58.43±3.57 | 46.51±4.52 | 78.28±3.83 | 71.03±3.48 |
| | AGC_IMVC | 80.17±3.19 | 70.86±6.14 | 67.79±4.88 | 52.41±5.92 | 83.79±3.83 | 76.03±4.54 |
| | LBIMVC | 78.79±3.02 | 74.66±3.69 | 70.31±1.95 | 65.82±2.72 | 85.17±0.94 | 84.27±2.70 |
| | HLSCG | 82.17±1.11 | 73.68±0.28 | 70.22±1.42 | 62.66±1.00 | 86.14±1.81 | 83.09±0.46 |
| | Proposed | **84.77±1.17** | **75.48±0.15** | **71.23±0.42** | **66.41±1.32** | **86.74±2.35** | **84.79±2.36** |
| COIL-20 | MVL_IV | 48.54±1.64 | 52.43±1.15 | 62.81±0.99 | 63.28±0.85 | 52.78±1.60 | 56.53±1.21 |
| | AWIMVC | 46.55±0.71 | 33.49±1.77 | 50.71±1.90 | 54.30±1.32 | 41.70±1.61 | 33.59±1.52 |
| | UEAF | 47.22±6.20 | 36.04±3.68 | 52.31±5.96 | 44.46±6.14 | 48.82±4.91 | 38.89±8.43 |
| | AGC_IMVC | 83.54±2.41 | 76.18±3.96 | 82.22±1.04 | 80.59±5.55 | 86.94±1.41 | 79.17±3.77 |
| | LBIMVC | 81.08±2.55 | 76.40±1.58 | 82.35±2.01 | 79.69±3.41 | 84.39±1.50 | 80.46±2.21 |
| | HLSCG | 82.27±2.06 | 78.85±1.32 | 84.63±1.38 | 78.60±1.43 | 85.50±2.44 | 76.90±2.08 |
| | Proposed | **84.05±2.46** | **80.65±3.40** | **85.64±3.10** | **80.06±1.79** | **87.72±2.57** | **80.64±1.72** |
| Caltech-7 | MVL_IV | 48.54±1.64 | 52.43±1.15 | 62.81±0.99 | 63.28±0.85 | 52.78±1.60 | 56.53±1.21 |
| | AWIMVC | 46.55±0.71 | 33.49±1.77 | 50.71±1.90 | 44.30±1.32 | 41.70±1.61 | 33.59±1.52 |
| | UEAF | 42.71±0.84 | 36.32±4.22 | 31.07±1.99 | 24.02±1.37 | 78.26±2.12 | 76.29±1.93 |
| | AGC_IMVC | 57.31±2.13 | 55.10±2.66 | 59.47±1.28 | 59.37±2.36 | 61.59±2.01 | 60.22±2.57 |
| | LBIMVC | 56.56±0.46 | 56.25±1.15 | 57.69±0.83 | 59.66±2.03 | 62.55±1.07 | 59.68±2.04 |
| | HLSCG | 65.13±1.52 | 63.45±2.21 | 64.77±2.31 | 61.21±1.75 | 79.50±1.29 | 77.36±1.90 |
| | Proposed | **66.73±2.35** | **64.38±3.24** | **67.06±2.31** | **63.49±2.37** | **80.46±2.29** | **78.55±2.62** |
| BUAA | MLV_IV | 36.81±1.16 | 31.11±0.27 | 66.83±2.53 | 62.63±0.14 | 38.16±1.01 | 32.74±0.41 |
| | AWIMVC | 41.02±1.67 | 33.27±0.68 | 62.09±1.05 | 61.22±0.93 | 43.68±0.48 | 36.49±0.74 |
| | UEAF | 30.59±3.97 | 26.74±9.66 | 60.46±7.94 | 59.14±3.38 | 32.22±4.41 | 28.07±3.97 |
| | AGC_IMVC | 26.74±0.67 | 25.48±2.70 | 53.87±0.61 | 57.51±5.25 | 28.01±1.37 | 26.67±3.72 |
| | LBIMVC | 46.49±2.63 | 39.65±1.60 | 58.56±0.50 | 54.33±3.75 | 39.45±2.73 | 37.60±2.01 |
| | HLSCG | 45.33±2.16 | 38.35±1.22 | 63.87±1.06 | 61.68±1.14 | 47.51±1.21 | 43.38±1.46 |
| | Proposed | **49.35±2.79** | **42.98±2.47** | **68.48±0.69** | **64.25±2.29** | **50.68±2.41** | **46.55±2.32** |
| Noisy COIL-20 | MVL_IV | 40.02±2.58 | 43.75±3.55 | 51.83±1.78 | 52.28±2.39 | 41.37±2.65 | 48.98±4.32 |
| | AWIMVC | 41.57±1.45 | 30.88±3.69 | 41.72±2.50 | 52.33±3.31 | 32.40±2.55 | 31.47±3.41 |
| | UEAF | 40.50±4.14 | 32.05±3.35 | 47.68±4.79 | 36.66±4.51 | 40.03±2.59 | 33.62±5.70 |
| | AGC_IMVC | 72.55±3.17 | 71.57±2.86 | 75.35±3.58 | 71.91±6.65 | 79.96±2.74 | 70.07±2.67 |
| | LBIMVC | 73.66±5.22 | 71.32±2.75 | 73.67±4.45 | 71.25±3.77 | 75.55±2.17 | 71.79±5.53 |
| | HLSCG | 71.12±2.87 | 69.82±3.11 | 71.55±1.20 | 70.46±2.55 | 81.04±1.54 | 72.31±3.32 |
| | Proposed | **75.24±3.31** | **73.88±2.70** | **78.33±2.71** | **75.34±1.80** | **83.25±4.64** | **74.55±2.36** |

the majority of the time cost is consumed by incorporating t-SVD, FFT, and inverse FFT operations, with respective computational complexities of $O(V^2 n^2)$ and $O(V n^2 log(n))$. Speciffcally, the computation of $J^{(v)}$, $R^{(v)}$, $M^{(v)}$, $F^{(v)}$, and $P^{(v)}$ only contain some simple operations, where the computational complexity can be ignored. Finally, the total time complexity can be calculated as $O(Vcn^2 + V^2 n^2 + V n^2 log(n))$.

## 3 EXPERIMENTS AND ANALYSIS

### 3.1 DATASETS DESCRIPTION AND INCOMPETED DATA CONSTRUCTION

**BBCSport**[1]**:** BBCSport constitutes a document database comprising 737 news articles pertaining to five sports: athletics, cricket, football, rugby, and tennis, sourced from the BBC Sport website between 2004 and 2005. For the purpose of evaluating various algorithms grounded in Integrated Multi-view Clustering (IMC), a subset from the BBC sport multi-view datasets, encompassing four distinct views, was utilized. This specific subset contains 116 samples, where each of the four views possesses a feature dimensionality of 1991, 2063, 2113, and 2158, respectively. **The Columbia**

---

[1]https://github.com/GPMVCDummy/GPMVC/tree/master/partialMV/PVC/recreateResults/data

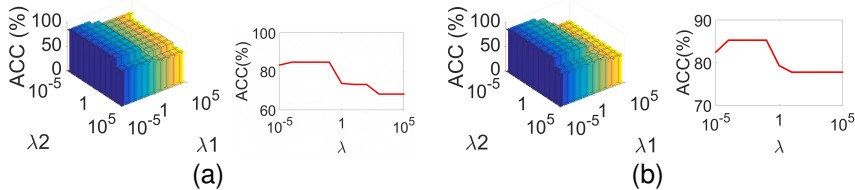

Figure 2: Clustering accuracies versus different combinations of parameters with 30% missing rate on the (a) BBCSport and (b) COIL-20databases, respectively.

**Object Image Library**[2] (COIL-20) comprises a total of 1,440 images spread across 20 different classes. Simultaneously, 10% Gaussian noise was imposed into the original samples to generate the Noisy COIL-20 database. To create the multi-view dataset, we extracted three feature types from each image: deep features using the VGG-F model Aiadi et al. (2024), Local Binary Patterns (LBP) features Zeng et al. (2023), and vectored pixel features. Meanwhile, the Caltech101 database encompasses 101 object categories, with each class having between 40 to 800 images Fei-Fei et al. (2004). For the comparison experiments conducted in this paper, a subset known as **Caltech-7** was chosen, which includes 1474 images from 7 classes. Specifically, the selected multi-view dataset for Caltech-7 consists of two views: GIST and LBP features Li et al. (2015). Referring to Zhao et al. (2016), a subset of the **BUAA-visnir face database** [3] was chosen to evaluate the proposed method, which comprises two views, i.e., 90 visual images and 90 near-infrared images, all belonging to the first 10 classes.

**Incomplete multi-view data construction:** In this paper, the incomplete multi-view dataset was created by randomly removing 30% and 50% instances in each view. Then, every algorithm method was carried out on each dataset 10 times, and the average value was reported as the final result. The evaluation metrics used were clustering accuracy (ACC), normalized mutual information (NMI), and purity Wen et al. (2021). All experiments conducted in this study were run on MATLAB R2020a, utilizing a hardware setup with 16.0 GB of RAM and a 3.40 GHz CPU.

### 3.2 COMPARISON WITH STATE-OF-THE-ART

In the comparative experiments, the proposed method was evaluated alongside a range of state-of-the-art IMC methods., i.e., AWIMVC Deng et al. (2020), UEAF Wen et al. (2019), AGC_IMVC Wen et al. (2021), LBIMVC Wen et al. (2024), and HLSCG Zhao et al. (2025). Table 1 reports the experimental results on the BBCSport, COLI-20, Caltech-7, and BUAA databases, respectively. From Table 1, it can be observed that: 1) The proposed method can always obtainsthe best clustering performances on different incomplete multi-view databases. It shows that graph-based methods have clear advantages compared to the other related methods. 2) Because the proposed method is capable of uncovering higher-order correlations among various views, it outperforms the other methods in terms of performance.

### 3.3 PARAMETER SENSITIVITY ANALYSIS

Algorithm 1 requires tuning of three key parameters, i.e., $\lambda_1$, $\lambda_2$, and $\lambda$. To determine the optimal parameter combinations for each dataset, we conducted extensive experiments on the BBC-Sport and COIL-20 datasets. As shown in Figure 2, our analysis reveals that the proposed method always demonstrates remarkable stability with respect to $\lambda_1$ across the range $[10^{-5}, 10^3]$ and $\lambda_2 \in [10^{-5}, 10^{-2}]$, as well as $\lambda$ in the range of $[10^{-5}, 10^{-2}]$.

### 3.4 CONVERGENCY ANALYSIS

We empirically validate the convergence of the proposed method through comprehensive experiments on the BBCSport and BUAA datasets. Figure 3 illustrates the relationship between iteration counts versus both clustering accuracy and objective function values across multiple datasets with

---

[2]http://www.cs.columbia.edu/CAVE/software/softlib/coil-20.php
[3]https://github.com/hdzhao/IMG/tree/master/data.

30% missing view rates. The results clearly demonstrate that clustering accuracies stabilize within few iterations while objective values decrease monotonically to convergence, thereby confirming the convergence properties of our method.

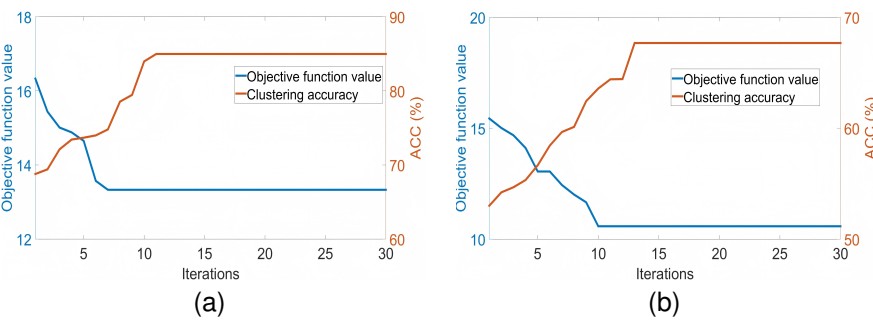

Figure 3: Objective function value and clustering accuracy of the proposed method versus the number of iterations on the (a) BBCSport and (b) BUAA databases, respectively.

## 3.5    ABLATION ANALYSIS

In this sub-section, experiments on the COIL-20 and noisy COIL-20 databses with 10%, 30%, and 50% view missing were conducted, where the proposed model with the high-confident block diagonal regularizer and the low-rank constraint were compared. Figure 4 shows the comparative results of the proposed method, where the proposed method signiffcantly outperforms the low-rank constraint model.

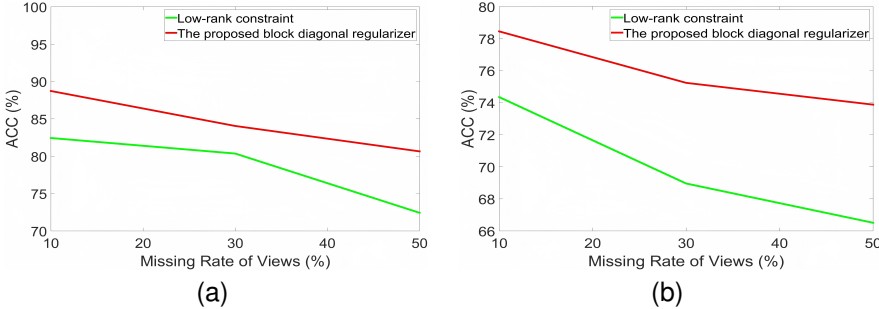

Figure 4: Clustering accuracies (ACC, %) of the proposed method and the conventional low-rank constraint model on the COIL-20 and noisy COIL-20 databases with different misng rates, respectively.

## 4    CONCLUSIONS

This paper proposed a novel and efficient CBDS_IMVC method for IMVC taks, which introduced a complete representation reconstruction strategy, where the consenus complete structure across all views was learned to preserve the structure of the reconstructed data. Specifically, a confident block diagonal regularizer is proposed to guide all views toward learning an invariable structure with block diagonal form. This mechanism preserves distribution of the data from each view while enforcing a consistent and strict block diagonal pattern that remains aligned across all views.Extensive experiments demonstrate superior missing data recovery and IMVC performance. Future work will extend this framework to broader application domains.

ACKNOWLEDGMENTS

This work was supported in part by the Natural Science Foundation of Guangdong under Grant 2024A1515011647, in part by Shenzhen Science and Technology Program under Grant JCYJ20240813105135047, and in part by the National Natural Science Foundation of China under Grant 62576112 and 62576140.

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
