# OpenReview forum: "Confident Block Diagonal Structure-Aware Invariable Graph Completion for Incomplete Multi-view Clustering"
_ICLR.cc/2026/Conference — ICLR 2026 Poster_

### Official Review · Reviewer_FAVd · 2025-10-27

**Soundness:** 2
**Presentation:** 2
**Contribution:** 2
**Rating:** 4
**Confidence:** 4

**Summary:**

This paper proposes an incomplete multi-view clustering method named CBDS_IMVC. By introducing confidence-aware block diagonal structure perception and invariant graph completion strategies, it aims to address the issue that existing methods overlook structural consistency and distribution differences during missing view recovery. The method jointly learns the complete representations of multiple views and employs block diagonal regularization to ensure all views share a consistent intrinsic clustering structure, thereby enhancing clustering performance. Experimental results demonstrate that this method outperforms state-of-the-art approaches on multiple benchmark datasets.

**Strengths:**

1. A confidence-aware block diagonal structure regularization is proposed to ensure that all views maintain a consistent local structure after recovery.

2. A joint learning framework is adopted to simultaneously optimize view recovery, graph completion, and clustering representation, thereby improving overall performance.

3. Multi-view complementary information is introduced to effectively leverage cross-view information for recovering the structure of missing instances.

**Weaknesses:**

1. While the empirical results are promising, a more in-depth theoretical analysis of why CBDS and invariable graph completion lead to better clustering performance would strengthen the paper.

2. Please double-check the definitions of variables, typos, and citations overall in the manuscript.

3. The complex mathematical solution process takes up too much space, resulting in insufficient necessary experiments, such as ablation experiments.

**Questions:**

1. How does the time complexity of the proposed method compare with other methods? Please supplement the analysis of time complexity.

2. Although the experimental results have demonstrated the superiority of this method, could you further explain the differences and connections between it and other block diagonal methods?

---

> ### Author Response · Authors · 2025-11-22
> **Responses**
>
> Dear Reviewer FAVd,
>
> We wish to express our sincere gratitude to the Reviewer for their thoughtful assessment and kind words regarding the presentation and analytical depth of our work. Their expert guidance has been instrumental in refining this manuscript. We have meticulously revised the paper to address all of the raised concerns. A detailed response to each specific point follows.
>
> AQ1: How does the time complexity of the proposed method compare with other methods?
>
> Responses: Many thanks for this comment. In Section 2.5 of the revised manuscript, we have added detailed time complexity analysis on the proposed method as “This paper employs an iterative optimization scheme to solve each variable in Problem (\cite{SO2}). The update procedures for $Y^{(v)}$ and $B$ involve only simple element-wise operations, rendering their computational cost negligible. The most expensive step is updating  $u$, which requires an eigenvalue decomposition of $O(cn^2)$ with the first  $c$ minimum eigenvalues.  In the step of $Z^{(v)}$, the majority of the time cost is consumed by incorporating t-SVD, FFT, and inverse FFT operations, with respective computational complexities of $O(V^2n^2)$ and $O(Vn^2log(n))$. Speciffcally, the computation of $J^(v)$, $R^{(v)}$, $M^{(v)}$, $F^{(v)}$, and $P^{(v)}$ only contain some simple operations, where the computational complexity can be ignored. Finally, the total time complexity can be calculated as $O(Vcn^2+V^2n^2+Vn^2log(n))$.”
>
> AQ2: Although the experimental results have demonstrated the superiority of this method, could you further explain the differences and connections between it and other block diagonal methods?
>
> Responses: Many thanks for this comment. In the second paragraph of Section 2.2, we have added more detailed description on the superiority of the proposed block diagonal regularizer as “By organizing multi-view similarities into a tensor, our model can natively account for high-order correlations and consensus among all views simultaneously. This provides a more comprehensive and coherent representation of the underlying data structure, leading to superior clustering performance, especially when views are complementary or heterogeneous.”.

---

> > ### Comment · Reviewer_FAVd · 2025-11-23
> >
> > The author has adequately addressed the issues I raised, so I have decided to increase the score of this paper.

---

### Official Review · Reviewer_vwi6 · 2025-10-28

**Soundness:** 3
**Presentation:** 4
**Contribution:** 3
**Rating:** 8
**Confidence:** 5

**Summary:**

This paper proposed a confident block diagonal structure-aware invariable graph completion-based incomplete multi-view clustering method, which designed a confident-aware missing-view inferring strategy to guarantee that recovered instances of all views have the same strict invariable local structure. This mechanism preserves distribution of the data from each view while enforcing a consistent and strict block diagonal pattern that remains aligned across all views.

**Strengths:**

1.This paper is well written with clear organization and significant contributions.

2.Different from the other related papers, this paper designed a complete representation reconstruction strategy. This approach learns a unified, consensus structure that is invariant across all views, effectively maintaining the intrinsic structure of the recovered data.

3.A novel confident block diagonal regularizer was proposed to guide all views toward learning an invariable structure with block diagonal form.

**Weaknesses:**

1.All equations presented in the manuscript must include clear definitions of their parameters.

2.Can the authors further clarify the importance of block diagonal structure learning compared with the other related methods.

3.The conclusion should clearly summarize the main findings.

**Questions:**

1.Compared with the other block diagonal regularizers, what are the obvious advantages of the proposed model?

2.Could the authors provide a more detailed explanation of Eq. (4)?

3.Could  the authors provide a more detailed explanation for the function of each part in Eq. (6)?

4.The mathematical formulas and explanations are too complex. Can the mathematical solution process be simplified, and add more explanations of the motivation and contributions of the proposed method?

---

> ### Author Response · Authors · 2025-11-22
> **Responses**
>
> Dear Reviewer vwi6,
>
> We extend our sincere appreciation to the Reviewer for their valuable feedback and generous comments on the clarity and analytical depth of our work. Their expert guidance was invaluable in revising the manuscript.
>
> AQ1: Compared with the other block diagonal regularizers, what are the obvious advantages of the proposed model?
>
> Responses: Many thanks for this comment. In the second paragraph of Section 2.2, we have added more detailed description on the superiority of the proposed block diagonal regularizer as “By organizing multi-view similarities into a tensor, our model can natively account for high-order correlations and consensus among all views simultaneously. This provides a more comprehensive and coherent representation of the underlying data structure, leading to superior clustering performance, especially when views are complementary or heterogeneous.”.
>
>
> AQ2: Could the authors provide a more detailed explanation of Eq. (4)?
>
> Responses: Many thanks for this comment. In the revised manuscript, we have added a more detailed explanation about Eq. (4) as “In multi-view learning, a fundamental assumption is that all views share a common, underlying data structure since they are generated from the same set of objects. When some views are incomplete, simply imputing missing data for each view independently can disrupt this shared structure, as the reconstructed data may no longer be consistent across views. ”.
>
> AQ3: Could the authors provide a more detailed explanation for the function of each part in Eq. (6)?
>
> Responses: Many thanks for this comment. In the final paragraph of Section 2.2, we have provided a more detailed explanation for the function of each part in Eq. (6) as “The first component is designed for missing data imputation, the second component is a block diagonal constraint, and the third component is responsible for maintaining structural consistency across different views.”
>
> AQ4: The mathematical formulas and explanations are too complex.
>
> Responses: Many thanks for this comment. We have simplified the entire optimization process by removing some complex and redundant mathematical formulas. The detailed optimization process can be seen in the uploaded manuscript.

---

### Official Review · Reviewer_uJCn · 2025-10-30

**Soundness:** 3
**Presentation:** 3
**Contribution:** 3
**Rating:** 4
**Confidence:** 5

**Summary:**

Incomplete multiview clustering is a significant and challenging sub-field within the research direction of multi-view clustering. This paper proposed a block diagonal structure completion method to guide the learning of the missing-instances of different views. Moreover, the re-constructed complete structures from different views can be adaptively aligned. The experimental results proved its positive performance in IMVC tasks.

**Strengths:**

1. The block-diagonal matrix represents a crucial property of data. It can clearly reveal the relationships within clusters and between clusters. This paper consider the IMVC problems from a unique perspective, where the property of the block diagonal structure of data is utilized to recover missing data from different views.

2. The proposed method designs the model simultaneously from the perspectives of missing data prediction and complete structure derivation. The joint learning strategy of this method enables the missing predicted data to maintain the structure of the original data on the one hand, and the complete predicted data to promote the derivation of the complete structure on the other hand.

**Weaknesses:**

1. The comparative experiments lack several state-of-the-art incomplete multi-view clustering (IMVC) methods. Including these would provide a more convincing evaluation of the proposed approach.

2. Could the authors provide further analysis or experimental evidence to justify the necessity of the proposed block diagonal constraint, particularly in comparison with self-representation or low-rank regularization techniques?

3. The paper would benefit from a more detailed explanation of the symbol “Pov” used in Equation (1), which would help readers better understand the proposed method.

4. It would be beneficial for the authors to incorporate additional visual evaluation approaches. Moreover, how did the authors set the hyperparameters? Please provide a complete set of basis and methods for setting them.

**Questions:**

See Weaknesses.

---

> ### Author Response · Authors · 2025-11-22
> **Responses**
>
> Dear Reviewer uJCn,
>
> We extend our sincere appreciation to the Reviewer for their valuable feedback and generous comments on the clarity and analytical depth of our work. Their expert guidance was invaluable in revising the manuscript.
>
> Q1. Add the state-of-the-art related methods in the comparative experiments.
> Responses: Thanks a lot for this comment. In the revised manuscript, we have added more state-of-the-art related methods in the comparative experiments, where Table 1 reported the experimental results.
>
> Q2: Could the authors provide further analysis or experimental evidence to justify the necessity of the proposed block diagonal constraint?
>
> Responses: Thanks a lot for this comment. In Section 3.5 of the revised manuscript, we have added the ablation experiments to prove the effectiveness of the proposed block diagonal constraint as “In this sub-section, experiments on the COIL-20 and noisy COIL-20 databses with 10%, 30%, and 50% view missing were conducted, where the proposed model with the high-confident block diagonal regularizer and the low-rank constraint were compared. Fig. 4 shows the comparative results of the proposed method, where the proposed method signiffcantly outperforms the low-rank constraint model.” Particularly, Figure 4 can be seen in the uploaded revised manuscript.
>
> Q3: The paper would benefit from a more detailed explanation of the symbol “Pov” used in Equation (1), which would help readers better understand the proposed method.
>
> Responses: Many thanks for this comment. In the first paragraph of Section 2.1, we have added more explanations on Pov as “Additionally,  $\mathcal{P}o_v$  denotes the projection matrix associated with view  $v$, which is used to record the existence of instances in the dataset  $X^{(v)}$. Specifically, it operates through one-hot encoding, where a value of 1 in a row signifies the presence of a specific instance in view $v$, while a value of 0 denotes its absence.

---

> > ### Comment · Reviewer_uJCn · 2025-11-27
> >
> > Thank you for your response. My concerns have been fully addressed, and I have therefore decided to raise my score.

---

### Official Review · Reviewer_o5ae · 2025-10-31

**Soundness:** 3
**Presentation:** 3
**Contribution:** 4
**Rating:** 6
**Confidence:** 5

**Summary:**

In this paper, a Confident Block-Diagonal Structure (CBDS)-based framework is presented for incomplete multi-view clustering. The key assumption is that the recovered instances across all views should exhibit a consistent and strictly invariant local structure, enforced through the CBDS constraint. To further enhance robustness, an invariable graph completion mechanism is introduced to infer the intrinsic cross-view structure while accommodating missing views. All components are jointly optimized in a unified learning framework, enabling mutual reinforcement between structure recovery and clustering. The experiments conducted on multiple benchmark datasets demonstrate that the proposed approach consistently outperforms baseline methods.

**Strengths:**

1. The method adopts a complete representation reconstruction mechanism to infer missing views, ensuring that the recovered multi-view representations preserve the underlying data structure across views.
2. By explicitly learning a consensus and complete structure shared by all views, the framework enhances global structure alignment and reinforces clustering consistency. This cross-view agreement helps achieve more stable and reliable clustering outcomes.
3. The introduction of a confident block-diagonal regularizer encourages each view to maintain its intrinsic distribution while enforcing a strict and invariant block-diagonal structure across views.

**Weaknesses:**

1. While the method handles missing views and distribution discrepancies, it does not explicitly address robustness to noise or outliers. It is better to include additional experiments with synthetic or real noise to demonstrate the method's resilience in practical scenarios.
2. The current Figure 1 provides only a high-level conceptual overview, and some steps in the pipeline remain unclear from the illustration. Please enhance the figure with clearer modules, detailed annotations, and flow directions.
3. The optimization part, especially Section 2.4, is a little long and detailed. It would be helpful to simplify this section or move some of the detailed derivations to the appendix.

**Questions:**

1. The transition from Eq. (6) to Eq. (7) feels a bit unclear and somewhat repetitive. Could this part be streamlined or explained more smoothly to make the flow easier for readers to follow?
2. Could you clarify why the element-wise Hadamard product is used in Eq. (6)? A bit more explanation on its purpose (e.g., handling observed entries or enforcing consistency) would help readers understand the motivation behind this choice.
3. Section 2.4 has a lot of detailed math, which may distract from the main ideas. Would it be possible to simplify this section or move some of the detailed derivations to the appendix so readers can focus more on the key concepts?
4. There are a few typos and formatting issues. Please proofread the paper to improve clarity.

---

> ### Author Response · Authors · 2025-11-22
> **Responses**
>
> Dear Reviewer o5ae,
>
> We wish to express our sincere gratitude to the Reviewer for their thoughtful assessment and kind words regarding the presentation and analytical depth of our work. Their expert guidance has been instrumental in refining this manuscript. We have meticulously revised the paper to address all of the raised concerns. A detailed response to each specific point follows.
>
> Q1. The robustness proof of the proposed method against noise.
> Responses: Many thanks for this suggestion. Aiming to evaluate the robustness of the proposed method, we have conducted the extended experiments on the noisy COIL-20 database with Gaussian noise. In Table 1 of the revised manuscript, it is obvious that the proposed method can achieve the highest accuracies.
>
> Q2. Enhance the descriptions on the caption of Figure 1.
> Responses: Thanks a lot for this comment. In the revised manuscript, we have added more detailed descriptions on the caption of Figure 1 as “The flowchart of the proposed CBDS_IMVC method. For the incomplete multi-view data, a complete representation reconstruction strategy is first designed to learn the consensus complete structure across all views. Afterward, we propose the confident block diagonal regularizer to guide all views toward learning an invariable structure with block diagonal form. Finally, the consensus representation can be learned for clustering.”
>
> Q3. Simplify the optimization process for the proposed manuscript in Section 2.4.
> Responses: Thanks a lot for this comment. Seeking to focus the reader's attention on the model itself, we have simplified the entire optimization process by removing some complex and redundant mathematical formulas. The optimized solution process can be seen in Section 2.4 of the revised manuscript.
>
> Q 4. The transition from Eq. (6) to Eq. (7) feels a bit unclear and somewhat repetitive.
> Responses: Thanks a lot for this comment. In the revised manuscript, we have added more detailed transition from Eq. (6) to Eq, (7) as “To complement the main objective and enhance the clustering performance, a consensus learning module is introduced to obtain a unified low-dimensional representation shared across all views for spectral clustering.”.
>
> Q5. Could you clarify why the element-wise Hadamard product is used in Eq. (6)?
> Responses: Many thanks for this comment. In the final paragraph of Section 2.2, we have added the explanation of the Hadamard product as “$\odot$ is the element-wise multiplication to preserve the real structure of the available instances”
>
> Q6. There are a few typos and formatting issues. Please proofread the paper to improve clarity.
> Responses: Thanks a lot for this comment. We have proofread the entire manuscript, and corrected all of the typos in the revised manuscript.

---

> > ### Comment · Reviewer_o5ae · 2025-11-25
> >
> > Thanks for the responses. Most of my concerns have been well-addressed, and I updated my score accordingly.

---

### Author Response · Authors · 2025-12-02
**Responses to ACs**

Dear Area Chairs,

Thanks a lot.

We sincerely thank the reviewers for their thoughtful comments on our work. Since all of the concerns from each reviewer were well tackled, the original scores before the rebuttal have been increased to positive accepting scores by the corresponding reviewers in the rebuttal.

Best regards

---

### Meta-Review · Area_Chair_bUa7 · 2026-01-06

**Summary:**

The paper proposes a confidence-aware block-diagonal structure regularization with invariable graph completion for incomplete multi-view clustering, and the reviewer–author discussion indicates the method is well-motivated and empirically competitive. So I recommend acceptance.

**Reviewer Concerns:**

The main concerns focused on (1) clarity and notation (e.g., definitions around projection matrices); (2) the need for stronger or more appropriate SOTA baselines; and (3) clearer evidence that the key block-diagonal constraints provide gains beyond simpler alternatives; these were addressed in the rebuttal via explicit clarifications and additional comparisons.

**Reviewer Scores:**

Overall sentiment trends positive, with at least one clear accept-level score and multiple reviewers explicitly indicating increased confidence and updated scores after the rebuttal, so the final recommendation is accept.

---

### Decision · Program_Chairs · 2026-01-26

Accept (Poster)